# UML-CoT: Structured Reasoning and Planning with Unified Modeling Language for Robotic Room Cleaning

## Abstract

Chain-of-Thought (CoT) prompting improves reasoning in large language models (LLMs), but its reliance on unstructured text limits interpretability and executability in embodied tasks. Prior work has explored structured CoTs using scene or logic graphs, yet these remain fundamentally limited: they model only low-order relations, lack constructs like inheritance or behavioral abstraction, and provide no standardized semantics for sequential or conditional planning. We propose **UML-CoT**, a structured reasoning and planning framework that leverages Unified Modeling Language (UML) to generate symbolic CoTs and executable action plans. UML class diagrams capture compositional object semantics, while activity diagrams model procedural control flow. Our three-stage training pipeline combines supervised fine-tuning with Group Relative Policy Optimization (GRPO), including reward learning from answer-only data. We evaluate UML-CoT on **MRoom-30k**, a new benchmark of cluttered room-cleaning scenarios. UML-CoT outperforms unstructured CoTs in interpretability, planning coherence, and execution success, highlighting UML as a more expressive and actionable structured reasoning formalism.

## 1 Introduction

Embodied AI systems, particularly those handling real-world tasks like robotic room cleaning, face significant challenges in multi-step reasoning and decision-making. These tasks require a nuanced understanding of object interactions, spatial relationships, and action dependencies. Recent advancements in large language models (LLMs) have enabled agents to generate reasoning traces through *Chain-of-Thought* (CoT) prompting (Wei et al., 2022; Kojima et al., 2022), leading to substantial improvements in planning and problem-solving. However, most existing CoT methods are limited by their reliance on unstructured, free-form text representations.

While **text-based CoT** approaches are flexible, they suffer from key limitations: (i) a lack of explicit structure to model object, action, and environment relationships, resulting in shallow world models (Zhang et al., 2024; Wang et al., 2023a), (ii) difficulty in interpreting or verifying reasoning traces, especially in cases with complex dependencies (Creswell et al., 2023), and (iii) susceptibility to ambiguity, repetition, and inconsistency across reasoning steps, which degrade planning quality and execution. To mitigate these issues, prior work has introduced symbolic representations like scene graphs or logic graphs (Pan et al., 2023; Besta et al., 2024; Yao et al., 2023) to bring structure into the reasoning process. Despite their advantages over plain text, **traditional graph-based reasoning** (Pan et al., 2023; Besta et al., 2024; Yao et al., 2023) has its own limitations. Graphs typically model binary or ternary relations but lack the expressive constructs needed for inheritance, aggregation, and behavioral abstraction. Furthermore, they lack standardized semantics for encoding procedural plans or control flows, making it challenging to represent sequential, conditional, or looping behaviors. Additionally, most graph-based methods are task-specific and require ad hoc modifications to accommodate new domains or reasoning levels.

To overcome these shortcomings, we adopt the Unified Modeling Language (UML)—a standardized formalism from software engineering (Ashbacher, 2004)—as the foundation for structured CoT reasoning and planning. UML addresses the deficiencies of graph-based reasoning: it complements

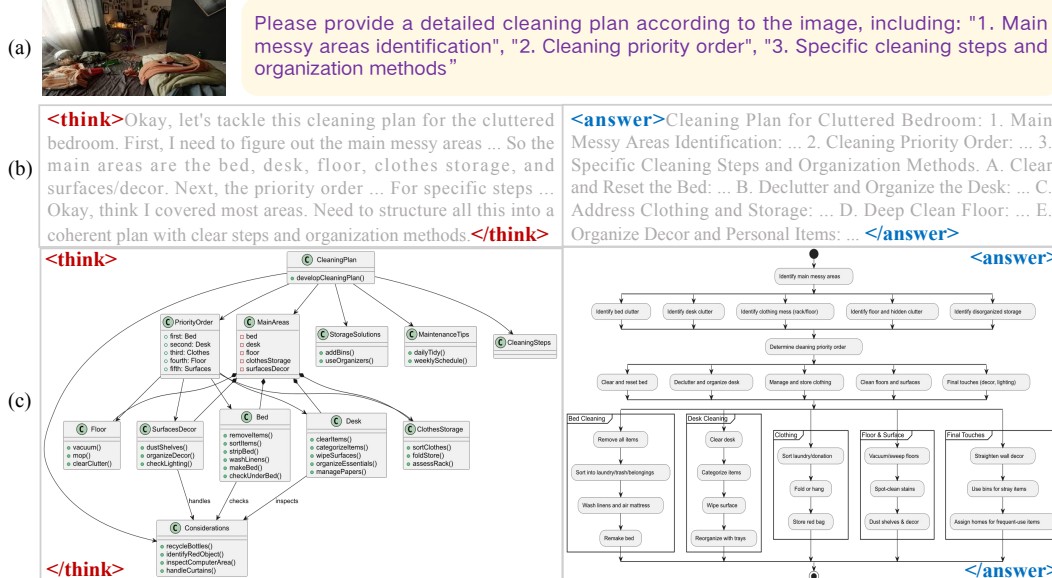

Figure 1: **Structured vs. Unstructured Chain-of-Thought Reasoning for Robotic Room Cleaning.** (a) Input: a cluttered room image and a text instruction. (b) Output from a plain-text CoT model, where reasoning and planning are expressed only in free-form language, lacking formal semantics and executable structure. (c) Output from our proposed UML-based framework, where reasoning is encoded as a UML class diagram and the corresponding plan is formalized as a UML activity diagram. This structured approach improves interpretability, ensures alignment between reasoning and action, and supports modular, executable planning. *Please zoom in to view details clearly.*

limited relational expressivity with class diagrams that natively model inheritance, aggregation, and object hierarchies; it resolves the absence of procedural semantics by providing activity diagrams for sequential, conditional, and iterative control flows; and it avoids task-specific fragmentation through a formalized syntax and semantics that ensure consistency and adaptability across domains. Additionally, UML's visual and modular nature enhances interpretability, making reasoning processes more transparent and verifiable in autonomous systems.

Building on these advantages, we introduce **UML-CoT**, a framework for structured reasoning and planning in embodied AI. The agent first performs symbolic reasoning over the environment by constructing UML class diagrams, representing objects, attributes, and relationships. It then generates an executable cleaning plan using UML activity diagrams, which describe sequential and conditional actions based on the physical scene. We introduce a **three-stage learning strategy** for training this framework: (1) Supervised fine-tuning (SFT) on annotated reasoning and planning traces in UML; (2) Reinforcement learning fine-tuning (RLFT) using Group Relative Policy Optimization (GRPO) (Shao et al., 2024), where the model receives rewards based on the correctness of the final plan; and (3) Further GRPO training on answer-only data to enable effective learning even without intermediate reasoning annotations. We evaluate our framework on the **MRoom-30k** dataset, which simulates diverse, cluttered room scenarios. Comparisons across four configurations—from plain-text reasoning to fully UML-based pipelines—show that our structured approach significantly improves plan coherence, execution success, and structural fidelity. Stage 2 RLFT further enhances performance, validating the effectiveness of staged reinforcement.

Recent embodied AI work, including SayCan (Ichter et al., 2022), RT-1/RT-2 (Brohan et al., 2023b;a), ThinkAct (Huang et al., 2025), and EMAC+ (Ao et al., 2025), has demonstrated the promise of combining LLMs with robotic planning. Yet these systems still rely on loosely structured intermediate forms. Our work shows that UML offers a standardized, interpretable, and domain-adaptable representation that improves reasoning fidelity and execution reliability. Fig. 1 illustrates the contrast between unstructured and UML-structured CoT reasoning in robotic room cleaning.

**Our contributions are:** (1) A UML-based structured reasoning and planning framework that unifies symbolic CoT reasoning with executable action planning for robotic room cleaning; (2) A three-stage training pipeline that combines supervised and reinforcement learning to optimize reasoning

quality and plan execution; (3) The introduction of **MRoom-30k**, a benchmark dataset of cluttered room scenarios for evaluating structured reasoning methods; (4) Empirical evidence that UML representations improve expressiveness, interpretability, and planning reliability compared to text-based and graph-based baselines.

## 2 RELATED WORK

**Chain-of-Thought Reasoning.** Chain-of-Thought (CoT) prompting enables large language models (LLMs) to perform multi-step reasoning by generating intermediate steps before the final answer (Wei et al., 2022). Variants such as Self-Consistency (Wang et al., 2023b) and Least-to-Most Prompting (Zhou et al., 2023) enhance robustness via sampling and decomposition strategies. Other improvements include iterative self-refinement (STaR (Zelikman et al., 2022)), debate-style prompting (ChainLM (Cheng et al., 2024)), and compressed intermediate reasoning (Compressed CoT (Cheng & Durme, 2024)). Several recent methods incorporate symbolic cues into CoT. Semi-Structured CoT (Su et al., 2024) blends structured graphs with unstructured context, while Faithful Logical CoT (Xu et al., 2024) and Structured CoT (Sultan et al., 2024) use logic programs or finite-state models. However, most of these approaches treat structure as auxiliary signals and operate on shallow or task-specific graphs. In contrast, our work frames CoT reasoning itself as a structured modeling task, using UML class diagrams to capture symbolic reasoning and UML activity diagrams to generate executable plans. This offers a unified, expressive, and interpretable framework that integrates reasoning and planning under a single formalism.

**Structured Reasoning and Symbolic Planning.** Graph-based symbolic reasoning has been explored to improve LLM interpretability and grounding. Scene graphs and logic graphs (Zhang, 2024; Xu et al., 2024) capture structured object-centric relations, while symbolic planners such as Sym-Planner (Xiong et al., 2025) enhance action generation through ranking or verification. Extensions of classical planning languages, including LLM+MAP (Chu et al., 2025) and InterPreT (Han et al., 2024), map natural language descriptions into PDDL for symbolic task planning. However, these approaches remain limited: they focus on basic relational modeling, lack constructs for procedural logic (e.g., conditionals, loops), and often require task-specific tailoring with poor generalization. In contrast, we adopt UML as a standardized and extensible formalism that integrates both reasoning and planning. UML class diagrams enable expressive structural modeling of objects, attributes, and hierarchies, while activity diagrams capture procedural control flows such as sequential, conditional, and iterative actions. Compared with PDDL's logic-centric syntax, UML provides richer expressiveness, standardized semantics, and visual interpretability, yielding a unified, transparent, and domain-adaptable symbolic interface from perception to action.

**Multi-Stage Training and GRPO.** Recent work has shown that combining supervised learning with reinforcement learning improves reasoning alignment and robustness. InstructGPT (Ouyang et al., 2022) and RRHF (Yuan et al., 2023) use reward-based fine-tuning to align outputs with human preferences. GRPO (Shao et al., 2024) introduces reward propagation to train intermediate reasoning steps based on final-answer feedback, improving CoT quality in low-supervision settings. Inspired by this, our framework adopts a three-stage pipeline tailored to structured CoT: (1) supervised fine-tuning on annotated UML diagrams; (2) intermediate RLFT using final-plan rewards to guide structural reasoning; and (3) answer-only GRPO to optimize planning when intermediate annotations are unavailable. This strategy improves both reasoning fidelity and execution success, especially in low-resource or semi-supervised scenarios

## 3 METHODOLOGY

### 3.1 TASK DEFINITION

We formulate room cleaning as a multimodal planning task, where an agent reasons about cluttered environments and generates executable cleaning strategies based on visual input. Each task consists of a single image $x$ of a messy room. The agent must produce two outputs: (i) a structured reasoning trajectory enclosed in <think>...</think> tags, and (ii) a final cleaning plan enclosed in <answer>...</answer> tags.

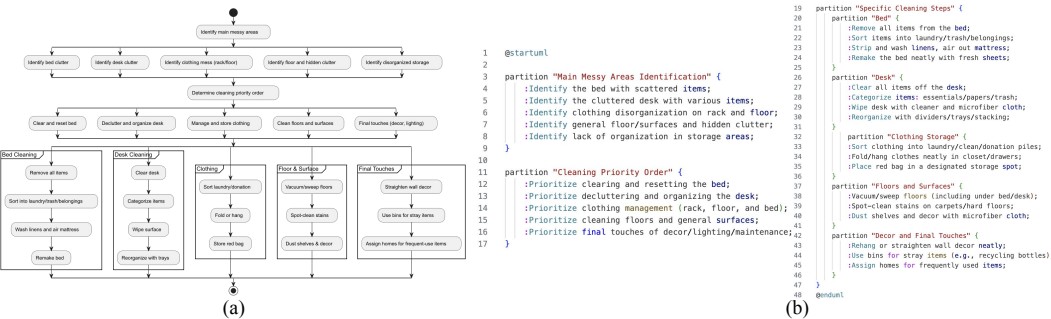

Figure 2: **UML class diagram and its corresponding PlantUML source shown in two equivalent forms:** (a) UML class diagram; (b) PlantUML source code. *Please zoom in to view details clearly.*

Figure 3: **UML activity diagram and its corresponding PlantUML source presented in two equivalent forms:** (a) UML activity diagram; (b) corresponding PlantUML code. *Please zoom in to view details clearly.*

Unlike conventional Chain-of-Thought (CoT) approaches using unstructured textual reasoning, we employ a symbolic representation based on Unified Modeling Language (UML):

- The reasoning process is represented as a **UML class diagram** $G_{\text{class}}$, capturing entities, attributes, and relationships.
- The cleaning plan is represented as a **UML activity diagram** $G_{\text{activity}}$, detailing an ordered sequence of actions with control-flow dependencies.

Formally, the objective is to learn a mapping: $f : x \rightarrow (G_{\text{class}}, G_{\text{activity}})$.

Note that the model generates PlantUML code, which can be rendered into UML diagrams using external tools. Figures 2 and 3 explicitly demonstrate this equivalence, showing that a UML diagram and its PlantUML source are two interchangeable representations of the same structure.

To evaluate plan quality, we define a semantic similarity metric between the predicted activity diagram $G_{\text{activity}}$ and its ground-truth reference, which serves as a reward signal in reinforcement learning. Details are provided in Section 3.4.

## 3.2 DATASET CONSTRUCTION

Existing indoor scene datasets, such as the MIT Indoor Scenes dataset (Quattoni & Torralba, 2009), suffer from a pronounced cleanliness bias—featuring predominantly tidy environments and lacking sufficient coverage of cluttered or disorganized household settings. To address this limitation, we conduct the **MRoom-30k** dataset, which focuses on messy indoor scenes for structured reasoning and cleaning plan generation. The dataset consists of 30,792 images sourced from Google, Bing, Baidu, and Rednote, as well as the Messy Rooms Dataset (Bhalgat et al., 2023). These images span various household environments (e.g., kitchens, bathrooms, bedrooms, and living rooms) and varying levels of messiness (mild, moderate, severe). MRoom-30k is annotated in two forms:

- **Standard Plans:** All images, except for a subset of 1,000, are annotated with final cleaning plans using GPT-4o, with consistent prompting to ensure a unified structure across outputs.

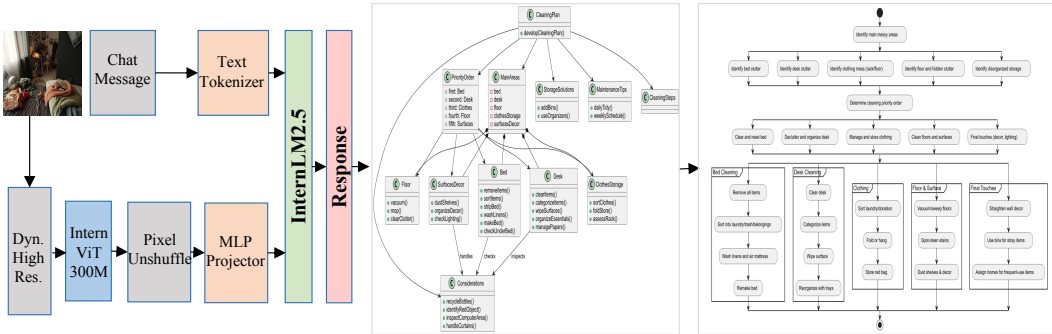

Figure 4: **Model architecture.** The image is preprocessed via dynamic resolution slicing and fed into a ViT-based encoder (InternViT-300M), followed by pixel unshuffling and MLP projection. The language decoder (InternLM2.5) receives both visual features and tokenized textual prompts, and generates two symbolic outputs: a UML class diagram for structured reasoning, and a UML activity diagram for executable cleaning plans. *Please zoom in to view details clearly.*

- **CoT-Enhanced Subset:** A random subset of 1,000 images is annotated with both intermediate reasoning (Chain-of-Thought, CoT) and final cleaning plans, generated using DeepSeek-R1.

Each instance is represented in both **textual** and **UML-based structured** formats for controlled comparison across different reasoning and planning modalities. Further details regarding the dataset construction and annotation process are provided in appendix A.

### 3.3 MODEL ARCHITECTURE AND I/O REPRESENTATION

We adopt **InternVL 2.5** (Chen et al., 2024; Wang et al., 2024) as the backbone for our structured multimodal reasoning framework. InternVL is a state-of-the-art vision-language model that integrates a visual encoder and a language decoder in a unified architecture, enabling effective grounding between image content and symbolic reasoning.

Each input instance consists of a single image depicting a cluttered room. To preserve both global context and local detail, InternVL apply a dynamic resolution slicing strategy that divides the image into fixed-size patches while retaining a resized global view. The visual features are passed through a pixel unshuffle and MLP projector before being fed into the decoder. Meanwhile, the text prompt is tokenized and also passed to the decoder, forming a joint multimodal input. The decoder component InternLM 2.5 then generates the symbolic reasoning and planning outputs. For per image, the model produces $G_{class}$ and $G_{activity}$.

This architecture (Fig. 4) enables joint modeling of visual perception and structured symbolic reasoning, producing interpretable outputs that bridge scene understanding and action generation.

### 3.4 MULTI-STAGE TRAINING STRATEGY

To equip the model with structured reasoning and planning capabilities in cleaning tasks, we propose a three-stage training strategy that progressively enhances performance through both supervised and reinforcement-based learning.

**Stage 1: Supervised Fine-tuning (SFT).** In Stage 1, we perform supervised fine-tuning to initialize the model's multimodal understanding and reasoning. Each instance consists of a room image and two structured outputs: (1) a UML class diagram in `<think>...</think>` tags, representing symbolic Chain-of-Thought (CoT) reasoning; (2) a UML activity diagram in `<answer>...</answer>` tags, encoding an executable cleaning plan grounded in CoT.

This stage trains the model to generate interpretable, structured outputs without explicit rewards, focusing on imitating high-quality reasoning and planning from annotated data. The success of later GRPO stages relies on the quality of this foundation, proved in section 4.4.

For subsequent experiments, we prepare variants with: (a) Plain-text CoT and plain-text plans, and (b) Plain-text CoT followed by UML-based plans. These variations allow us to analyze the impact of different reasoning and output structures.

**Stage 2: Reinforcement Learning Fine-tuning (RLFT)** In Stage 2, we apply Reinforcement Learning Fine-tuning (RLFT) on the same CoT-annotated dataset used in SFT. The reward is computed **only based on the final answer**, enabling indirect supervision of intermediate reasoning. After the SFT stage, the model is capable of correctly outputting format tags, class diagrams, and activity diagrams, which are prerequisites for RLFT.

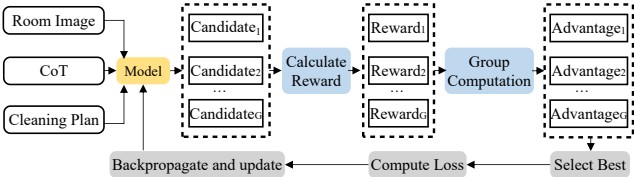

Figure 5: **Overview of Group Relative Policy Optimization (GRPO)**, applicable to both Stage 2 and Stage 3. The model receives image, CoT, and plan as input, generates multiple candidates, evaluates them using advantage scores, and updates its parameters based on the best-scoring candidate.

As shown in Fig. 5, the model receives three inputs: a room image, a Chain-of-Thought (CoT), and the corresponding ground-truth cleaning plan. It generates $G$ plan candidates, and for each, a composite reward is computed as:

$$\text{reward} = \text{format\_reward} + \text{accuracy\_reward} \tag{1}$$

- **Format Reward:** 1.0 if both `<think>` and `<answer>` tags are present; 0 otherwise.
- **Accuracy Reward:** Based on semantic similarity between the predicted and reference UML activity diagrams, computed using `all-MiniLM-L12-v2`.

The diagram is parsed into three predefined partitions: *Main Messy Areas*, *Cleaning Priority Order*, and *Specific Cleaning Steps*. A greedy matching algorithm is used to compute the accuracy reward based on cosine similarity between predicted and reference nodes.

The raw rewards for all $G$ candidates are normalized to obtain a relative advantage score:

$$\text{advantage}_i = \frac{r_i - \mu}{\sigma + \epsilon} \tag{2}$$

where $\mu$ and $\sigma$ are the mean and standard deviation of rewards among the candidates, and $\epsilon$ is a small constant for stability. The candidate with the highest advantage score is selected, and its log-probability is scaled by the advantage score. The final policy loss is:

$$\mathcal{L} = -\log \pi_\theta \left( \hat{y} | x \right) \cdot \text{advantage} \left( \hat{y} \right) \tag{3}$$

This loss is backpropagated to update the model. Notably, although CoT is part of the input, rewards are computed only on the final answer, allowing the model to refine its reasoning chain through latent reward propagation.

Since only structured UML outputs are used at this stage, discussion of purely textual rewards is excluded in this phase.

**Stage 3: Guided Reward Propagation Optimization (GRPO).** In Stage 3, we apply GRPO to a broader dataset with only final cleaning plans annotated. The model receives a room image and a ground-truth plan as input, without chain-of-thought reasoning traces, and generates $G$ candidate plans. Rewards are computed (Equation 1) for each candidate to guide model updates.

For reward evaluation, two pipelines are used: **For UML-based outputs**, the reward function from Stage 2 is reused, including UML verification and semantic similarity across structured partitions. **For textual outputs**, a simplified reward is used:

- **Format reward**: 1.0 if both `<think>` and `<answer>` tags are present; 0 otherwise.

- **Accuracy reward**: Cosine similarity between predicted and reference plans, using `all-MiniLM-L12-v2` embeddings.

After computing raw rewards, group normalization is applied to obtain advantage scores (Equation 2). The candidate with the highest advantage score is selected, and the final loss is computed (Equation 3), allowing the model to improve plan quality through reward-based fine-tuning, even without intermediate reasoning supervision.

## 4 EXPERIMENTS

### 4.1 EXPERIMENTAL SETUP

**Dataset.** We conduct our experiments on MRoom-30k, introduced in section 3.2. Except 1,000 instances annotated with reasoning traces, the remaining samples are randomly split into 80% for training, 10% for validation, and 10% for testing. Due to computational constraints, 2,000 training samples are randomly selected for Stage 3 GRPO fine-tuning. During evaluation, we sample 1,000 test instances to assess model performance across all metrics.

**Implementation Details.** The model backbone is based on InternVL 2.5-1B. We investigate four distinct input-output configurations: (i) Textual CoT → Textual Cleaning Plan, (ii) Textual CoT → UML-based Cleaning Plan, (iii) UML-based CoT → UML-based Cleaning Plan, and (iv) UML-based CoT → UML-based Cleaning Plan (3-stage). The first configuration, Textual CoT → Textual Cleaning Plan, is a well-established approach found in VLM-R1 (Shen et al., 2025). In addition to this, we compare the performance of Tree of Thoughts (ToT) (Yao et al., 2023) and Graph of Thoughts (GoT) (Besta et al., 2024) against our proposed configurations. Complete training arguments across three stages are shown in appendix C.

**Evaluation Metrics.** Conventional metrics such as ROUGE are inadequate for assessing room-cleaning plans due to their subjective nature. We therefore adopt a semantic similarity–based evaluation pipeline using the `all-MiniLM-L12-v2` model. Predictions and references are decomposed into structured partitions, and node-level alignment is performed via similarity matrices with greedy matching. We report both regression-style metrics, defined as the average similarity across all matched node pairs, and classification-style metrics, where a fixed threshold (0.5) determines matches: nodes matched above the threshold are TP, unmatched ground-truth nodes are FN, and unmatched predictions are FP. Precision, Recall, and F1-score are then computed from these counts.

In particular, we interpret **Recall as the task execution success rate**, as it measures the proportion of ground-truth steps covered by the model's prediction. High recall ensures that critical cleaning steps are included, whereas lower precision (due to redundant steps) may still yield successful execution. To ensure consistent evaluation, textual instructions are converted into UML activity diagrams using GPT-4o before scoring.

### 4.2 TRAINING DYNAMICS

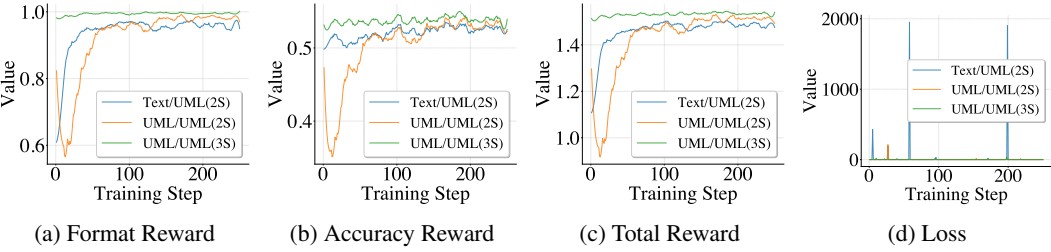

(a) Format Reward      (b) Accuracy Reward      (c) Total Reward      (d) Loss

Figure 6: **Comparison of training metrics across experimental configurations:** (a) Format reward, (b) Accuracy reward, (c) Total reward, and (d) Loss.

We compare three configurations during Stage 3 GRPO fine-tuning: i) Textual CoT → UML answer (w/o Stage 2); ii) UML CoT → UML answer (w/o Stage 2); and iii) UML CoT → UML answer (w/ Stage 2). Since the text-based configuration uses a purely textual dataset and converts outputs to

UML format only during evaluation, its training dynamics are indirectly reflected through post-hoc conversion. For fair comparison, we omit its learning trends in this section.

**Effect of Structured Reasoning.** As shown in Fig. 6a–6c, the two-stage model using UML-based CoT (UML/UML 2S) initially trails but gradually catches up even slightly outperformss its textual CoT counterpart (Text/UML 2S) across all reward metrics. Specifically, UML-based reasoning achieves significantly higher format reward due to better syntactic consistency, and also surpasses textual reasoning in semantic alignment and total reward.

This performance gap highlights the advantage of structured intermediate representations. Even without additional reinforcement optimization, UML-based CoT enables more interpretable and accurate planning compared to unstructured textual chains, confirming the benefits of symbolic abstraction.

**Impact of Intermediate RLFT.** Further gains are achieved by introducing an additional intermediate RLFT stage. The three-stage model (UML/UML 3S) not only achieves the highest overall reward scores, but also shows smoother and more stable convergence behavior, as seen in Fig. 6d. In contrast, the two-stage variants exhibit larger fluctuations and higher loss variance, especially in the textual setting.

This improvement validates the role of Stage 2 RLFT as a targeted optimization step: by leveraging reward signals from final plans, it indirectly refines the reasoning process, leading to better planning quality and training stability. These findings support our design of progressive reward-guided optimization over structured reasoning.

## 4.3 EVALUATION RESULTS

Beyond training dynamics, we also benchmark our method against state-of-the-art approaches—including text-based CoTs (Shen et al., 2025), Tree of Thoughts (Yao et al., 2023), and Graph of Thoughts (Besta et al., 2024)—with results summarized in Table 1.

Table 1: **Performance comparison with state-of-the-art approaches.** Best results are highlighted in bold, and second-best results are underlined.

| | Method | Similarity | Precision | Recall | F1 | Success Rate |
|---|---|---|---|---|---|---|
| SOTA | Tree of Thoughts (Yao et al., 2023) | 0.4209 | 0.4854 | 0.4639 | 0.4695 | 0.4639 |
| | Graph of Thoughts (Besta et al., 2024) | 0.5383 | 0.5263 | 0.5579 | 0.5371 | 0.5579 |
| | Text/Text (2S) (Shen et al., 2025) | 0.5498 | **0.5489** | 0.6280 | 0.5811 | 0.6280 |
| Ours | Text/UML (2S) | 0.5562 | 0.5384 | 0.6438 | 0.5812 | 0.6438 |
| | UML/UML (2S) | 0.5617 | 0.5304 | 0.6536 | 0.5803 | 0.6536 |
| | UML/UML (3S) | **0.5694** | 0.5326 | **0.6744** | **0.5904** | **0.6744** |

Notably, while Textual CoT with Textual Plan achieves the highest precision (0.5489), it lags behind in recall and overall F1, suggesting that while it generates more accurate matches, it fails to capture a significant portion of valid responses. The Tree of Thoughts and Graph of Thoughts demonstrate lower performance across key metrics, particularly in similarity and recall, despite achieving relatively higher precision.

By contrast, switching the plan output from text to UML (Text/UML 2S) results in improvements in similarity and recall, indicating that UML-based outputs facilitate better alignment with structured targets, even when the CoT remains textual. Furthermore, When both CoT and plan are represented as UML (UML/UML 2S), the model further benefits in recall and similarity, validating the effectiveness of fully symbolic reasoning.

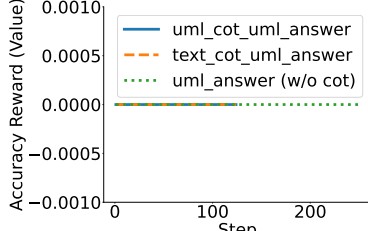

Figure 7: **Comparison of accuracy reward across different datasets.**

Among all settings, the best overall performance is achieved by the UML/UML (3S) configuration, with the highest semantic similarity (0.5694), recall (0.6744), and F1 score (0.5904), demonstrating

that both structured representation and additional RLFT fine-tuning in Stage 2 (even without inter-midiate reasoning supervision) contribute significantly to improved planning quality.

## 4.4 ABLATION STUDY

**Necessity of SFT Prior to GRPO**   To evaluate the necessity of SFT, we tested whether GRPO alone could generate valid UML outputs by applying it to a dataset containing only annotated final answers. Since the reward function relies on detecting UML tags and calculating accuracy, models without prior SFT failed to produce UML-formatted outputs, resulting in zero accuracy rewards and undefined advantage functions. We also applied GRPO to datasets with UML- and text-based reasoning, but observed the same issue, as rewards were only given for final answers. These results, shown in Fig. 7, highlight that SFT is crucial for GRPO to effectively generate UML-formatted answers.

**Effect of GRPO beyond SFT**   To determine whether performance gains arise from longer training or Group Relative Policy Optimization (GRPO), we analyze test results under different SFT epochs. As shown in Fig. 8, similarity, precision, recall, and F1 improve rapidly in the first few epochs but plateau after the 5th epoch, indicating convergence. In contrast, applying GRPO from the SFT epoch 5 checkpoint results in consistent improvements across all metrics according to Table 1. Since extended SFT alone does not produce these gains, we attribute the improvements to GRPO's reward-based optimization. This comparison highlights that i) SFT saturates around 5 epochs, and ii) GRPO provides additional improvements.

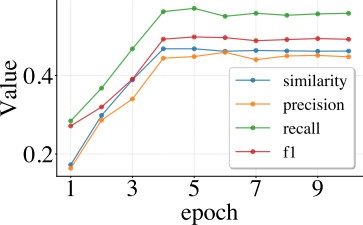

Figure 8: **Test set metrics across different SFT epochs.**

## 4.5 CROSS-TASK GENERALIZATION

Table 2: **Generalization Results.**

| Task | Model | Similarity | Precision | Recall | F1 |
|------|-------|-----------|-----------|--------|-----|
| Cooking | text/text 2S | 0.6357 | 0.2010 | 0.4219 | 0.2694 |
|  | text/uml 2S | 0.5705 | 0.2683 | 0.4269 | 0.3213 |
|  | uml/uml 2S | 0.6058 | 0.4119 | 0.4059 | 0.3076 |
|  | uml/uml 3S | 0.6889 | 0.3447 | 0.4448 | 0.3824 |
| Painting | text/text 2S | 0.6040 | 0.1471 | 0.3665 | 0.2087 |
|  | text/uml 2S | 0.5750 | 0.1750 | 0.1555 | 0.1643 |
|  | uml/uml 2S | 0.6156 | 0.1566 | 0.1498 | 0.1531 |
|  | uml/uml 3S | 0.6503 | 0.1892 | 0.2769 | 0.1715 |

Cross-task generalization is crucial for evaluating how well models can adapt to different tasks beyond their training domain. Table 2 evaluates model performance across cooking and painting tasks. In cooking, the 'uml/uml 3S' model performed best, particularly in similarity and F1 score, suggesting strong task generalization. In contrast, the 'text/text 2S' model, despite better precision and recall, underperformed in F1, highlighting the limitations of text-based models. For painting, all models showed poor performance, with 'uml/uml 3S' slightly outperforming others in similarity but still struggling in F1. This emphasizes the challenge of cross-task generalization, indicating that further task-specific model improvements are necessary.

## 5 CONCLUSION

We present a structured reasoning framework for multimodal room cleaning that integrates UML-based representations into the Chain-of-Thought (CoT) paradigm. Reasoning is expressed as UML class diagrams and executable plans as UML activity diagrams, yielding an interpretable pipeline grounded in visual input. A progressive three-stage training strategy—SFT, RLFT, and GRPO—further refines reasoning and planning. Experiments on MRoom-30k show that UML-based CoT surpasses textual and mixed baselines, while GRPO improves structural fidelity and execution quality. Semantic similarity–based evaluation confirms the robustness and effectiveness of our approach.

## ETHICS STATEMENT

Our work adheres to ethical standards in data collection and usage. The MRoom-30k dataset consists of publicly available images collected from platforms such as Google, Bing, Baidu, and Xiaohongshu. These images were carefully curated to ensure compliance with platform privacy policies. No personal or sensitive data is involved, and the dataset does not contain identifiable information. All image annotations and subsequent model training were conducted with a focus on fairness and minimizing biases. Ethical considerations, particularly around the use of AI for real-world tasks such as room cleaning, were rigorously assessed to ensure the safety, transparency, and fairness of the models employed.

## REPRODUCIBILITY STATEMENT

The models, datasets, and code used in this research will be made publicly available in the near future. Once released, detailed instructions for replicating the experiments will be provided, including the dataset, environment setup, and evaluation metrics. We are committed to ensuring that the research is reproducible and accessible to the broader research community for validation and further exploration.

## LLM CLARIFICATION

In this research, GPT-4o was used for data annotation, specifically for generating textual cleaning plan and PlantUML cleaning plan associated with the images in the MRoom-30k dataset. Additionally, GPT-4o was utilized during the evaluation phase to convert textual plans into UML activity diagrams. Furthermore, DeepSeek is employed for annotating Chain-of-Thought (CoT) data, enabling structured reasoning traces to guide the task.

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

## A MRoom-30k Dataset Construction

This section provides a detailed description of the construction and annotation of the MRoom-30k dataset, a large-scale visual dataset of messy indoor scenes used for training our structured reasoning agent.

### A.1 Data Collection and Cleaning

To address the clean-scene bias of existing indoor datasets (e.g., MIT Indoor Scenes), we curate a large-scale image dataset focusing on cluttered and disorganized household environments. MRoom-30k contains 30,792 high-quality images collected from four platforms: Google, Bing, Baidu, and Xiaohongshu (RED). These images cover a wide variety of spaces including kitchens, bathrooms, bedrooms, living rooms, balconies, and garages, and span varying levels of messiness (mild, moderate, severe).

**Multi-Platform Crawling Strategy.** We designed platform-specific query strategies to reflect linguistic, cultural, and platform-dependent nuances. A total of 100 search queries were used (25 per platform). Western platforms like Google and Bing emphasize metaphorical or event-driven expressions (e.g., "post-party dorm mess"), while Baidu queries target Chinese lifestyle scenarios, and Xiaohongshu queries incorporate younger-generation and emotionally expressive hashtags. These queries span seven scene categories and seven messiness types. Full query lists are given in Table 3.

| Google | Bing | Baidu (translated) | Xiaohongshu (translated) |
|---|---|---|---|
| Hoarding disorder room | Trash-filled apartment | Real photos of dirty rental rooms | Rental warning: messy rooms in real life |
| Post-party messy dorm | Abandoned squatter house | Greasy kitchens in old apartments | Student dorms after chaos |
| Cluttered desk with food scraps | Dirty mattress on floor | Failed waste sorting scenes | Messy room self-rescue for loners |
| Moldy bathroom corners | Pet hair-covered couch | Failed dorm hygiene inspections | Hygiene issues in co-living apartments |
| Overflowing garbage can | Decomposing food pile | Landlord's nightmare tenant rooms | Influencer Airbnb fails |
| Greasy kitchen cabinets | Tornado-hit kids playroom | Mold stains in bathrooms | Trash house before renovation |
| Pile of unwashed dishes | Trashed living room | Living rooms piled with junk | Cleaners' breakdown moments |
| Broken furniture clutter | Spring cleaning failure | Balconies used as recycling depots | Scary scenes left after moving out |
| Cockroach-infested kitchen | Airless moldy basement | Expired food mold in fridges | Room conditions after lease termination |
| Stained carpet close-up | Bachelor pad disaster | Dust and spider webs under bed | Pet disaster aftermath |
| Rotting food in fridge | Post-riot room chaos | Trash-filled bedrooms | Makeup spilled all over desk |
| Post-apocalyptic room | Squalid homeless shelter | Extremely messy bedrooms | Instant noodle soup-soaked carpet |
| Pigsty-like bedroom | Stained mattress dump | Cockroach nests | Expired moldy cosmetics |
| Disaster zone kids room | Cigarette butt-filled ashtray | Smelly and disgusting toilets | Living room buried in delivery boxes |
| Biohazard-level bathroom | Zombie apocalypse bedroom | Floor too dirty to step on | Mold spots from wet laundry |
| Post-gaming session mess | Overflowing diaper pail | Post-quarantine room scenes | OCD warning: extreme mess |
| Moldy refrigerator interior | Broken glass and debris | Hygiene disputes in co-rentals | Ghosting friends due to mess |
| Clogged sink with sludge | Dorm room after finals | Food boxes growing worms | Landlord confiscating deposit scenes |
| Depression nest reality | Cluttered makeup vanity | Moldy secondhand furniture | Students after finals chaos |
| College frat house filth | Disaster area garage | Wall leakage and mildew | One-month quarantine without cleaning |
| Moldy walls in rainy season | Peeling wallpaper mold | Water dripping from walls in humid season | Slightly dusty windowsills |
| Heater leak damaged floor | Shoes scattered at entrance | Flooded room from northern heater bursts | Mountain of delivery boxes from sales |
| Dorm move-out day mess | Dusty bookshelf neglect | Junk-filled hallways in old apartments | Shocked landlord on move-out day |
| Pet urine stained carpet | Cluttered garage with tools | Battery fire hazard in shared hallway | Urine-stained carpet cry for help |
| Thanksgiving party aftermath | Christmas decoration chaos | Kitchen chaos after New Year's Eve dinner | Moldy clothes in wardrobe during rainy season |

Table 3: Cross-Platform Crawling Keywords for Messy Room Image Collection

**Crawler Architecture and Anti-Bot Strategy.** We implement a three-stage multi-engine crawler covering Google, Bing, and Baidu. The pipeline includes query URL generation, page traversal, and image parsing. As shown in Figure 9, the system dynamically switches between Selenium-based browser simulation (for JavaScript-driven engines like Google and Bing) and API-based access (for Baidu). Anti-crawling countermeasures include randomized headers, proxy rotation, scroll simulation, and delayed thumbnail expansion.

For Xiaohongshu, which is login-gated, we design a signed API pipeline using pre-acquired cookies and encrypted signature tokens (X-s and X-t). Multi-stage signing and response token decoding allow for scalable retrieval of public note images.

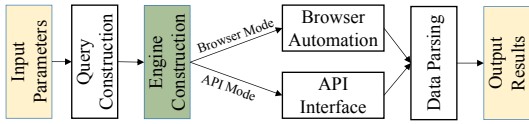

Figure 9: Crawling System Architecture

## A.2 DATA FILTERING AND DEDUPLICATION

Due to duplicate URLs, CDN variations, and repeated crawling attempts, redundant and near-duplicate images frequently occur. To ensure training quality and generalization, we apply two-layer filtering:

**(1) File-level Hashing.** We compute MD5 fingerprints to remove byte-identical images.

**(2) Perceptual Hashing (pHash).** We apply perceptual hashing based on DCT coefficients of grayscale-resized images. Images with Hamming distances below a threshold are considered visually redundant.

**(3) Irrelevant Image Filtering.** To detect irrelevant or clean images (e.g., text-only slides, tidy apartments), we prompt InternVL 2.5-38B with: "`Is this a messy room? Answer with only yes or no.`" Images with negative answers are excluded from the dataset.

After filtering, we obtain a curated dataset of 30,792 high-quality messy room images.

## A.3 ANNOTATION PIPELINE

We annotate MRoom-30k with structured cleaning plans to support reasoning-based model training. It should be noted that for both base plan annotation and chain-of-thought annotation, we prepare textual annotation and UML-based annotation.

**Base Plan Annotation (30k).** We prompt chatgpt-4o to generate detailed cleaning plans including three structured fields: (i) messy area identification, (ii) cleaning priority, and (iii) step-by-step actions.

Prompt used:

> *Please provide a detailed cleaning plan according to the image, including:*
> *1. Main messy areas identification*
> *2. Cleaning priority order*
> *3. Specific cleaning steps and organization methods*

**Chain-of-Thought Annotation (1k).** We select 1,000 images from MRoom-30k for Chain-of-Thought (CoT) annotation. As DeepSeek-R1 is not multimodal, we first use InternVL 2.5 to extract image descriptions. These descriptions are then passed to DeepSeek Reasoner via API to generate a reasoning trajectory and final cleaning plan.

**Annotation format conversion.** While the basic cleaning plan annotations and chain-of-thought annotations introduced earlier are originally provided in textual form, subsequent experiments require them to be represented in UML format to ensure structural consistency and enable symbolic reasoning. To accomplish this conversion, we employ ChatGPT-4o with the following prompt:

- Chain-of-Thought:

    Convert the following structured plan into a PlantUML activity diagram. The output should only include three partitions: "Main Messy Areas Identification","Cleaning Priority Orde", and"Specific Cleaning Steps". Return only valid PlantUML code, starting with @startuml and ending with @enduml.

- Cleaning Plan:

    Convert the following thinging process into a PlantUML class diagram. Return only valid PlantUML code, starting with @startuml and ending with @enduml.

This completes the construction of the MRoom-30k dataset and its structured annotations.

---

**Algorithm 1** Reward Computation Pipeline

---

**Require:** Predicted output $y_{\text{pred}}$, reference output $y_{\text{ref}}$
1: Initialize format_reward $\leftarrow 0$, accuracy_reward $\leftarrow 0$
2: **if** $y_{\text{pred}}$ contains valid tags `<think>` and `<answer>` **then**
3:     format_reward $\leftarrow 1.0$
4: **end if**
5: **if** reference output is UML **then**
6:     **if** $y_{\text{pred}}$ contains `@startuml` and `@enduml` **then**
7:         **for all** partition $p \in \{\text{MessyAreas}, \text{PriorityOrder}, \text{Steps}\}$ **do**
8:             **if** $p$ exists in both $y_{\text{pred}}$ and $y_{\text{ref}}$ **then**
9:                 Encode all nodes in $p$ using MiniLM $\rightarrow$ vectors
10:                Compute similarity matrix $S_p$
11:                Perform greedy node matching in $S_p$
12:                Accumulate similarities for matched pairs
13:             **else**
14:                Add $0.0$ for all unmatched ground-truth nodes in $p$
15:             **end if**
16:         **end for**
17:         accuracy_reward $\leftarrow$ average of all matched node similarities
18:     **else**
19:         accuracy_reward $\leftarrow 0.0$
20:     **end if**
21: **else if** reference output is text **then**
22:     Encode $y_{\text{pred}}$ and $y_{\text{ref}}$ as whole paragraphs
23:     accuracy_reward $\leftarrow \cos(v^{\text{pred}}, v^{\text{ref}})$
24: **end if**
25: **return** reward = format_reward + accuracy_reward

---

## B   REWARD COMPUTATION PIPELINE

To optimize the model using Group Relative Policy Optimization (GRPO), we design a reward pipeline that evaluates the quality of generated cleaning plans through two components: syntactic validity and semantic correctness. The full logic for reward evaluation—including format checking, semantic comparison, and routing—is detailed in Algorithm 1. The total reward is defined as:

$$\text{reward} = \text{format\_reward} + \text{accuracy\_reward} \tag{4}$$

**1. Format Reward.** This binary reward evaluates syntactic correctness:

$$\text{format\_reward} = \begin{cases} 1.0 & \text{if output passes format checks} \\ 0.0 & \text{otherwise} \end{cases} \tag{5}$$

We consider format valid if the output includes both `<think>` and `<answer>` blocks.

**2. Accuracy Reward.** The accuracy reward measures the semantic quality of a generated plan relative to the reference plan. We support two formats:

- **UML-based plan evaluation**: used when the output is a structured UML activity diagram.

- **Text-based plan evaluation**: used when the output is a plain-text cleaning plan.

(A) UML-BASED EVALUATION. The UML-based accuracy reward begins by verifying whether the generated output includes valid PlantUML syntax markers (`@startuml` and `@enduml`). Only outputs that pass this check are eligible for further structural evaluation. If the output lack PlantUML syntax markers, the accuracy reward is 0.

Following the Stage 1 SFT prompt design, all activity diagrams are expected to follow a standardized three-part structure: (i) **Main Messy Areas Identification**, (ii) **Cleaning Priority Order**, and (iii) **Specific Cleaning Steps**

To robustly extract the content of each partition, we implement a stack-based bracket matching algorithm. This approach uses a nesting counter to ensure that all sub-activity nodes are fully and correctly parsed within their respective partitions.

Unlike traditional string-matching approaches, we use semantic similarity to compare nodes. Activity nodes such as "organize the desk surface" and "clear items from the desk" may differ lexically but are semantically equivalent. To capture this, we employ the `all-MiniLM-L12-v2` from Sentence Transformer to encode each node into a semantic vector.

For each of the three partitions:

- If a partition is missing in the prediction, all ground-truth nodes under that partition are considered unmatched (assigned similarity 0.0).

- If both prediction and reference contain the partition, we compute pairwise cosine similarities between all activity nodes in that partition.

A greedy matching algorithm is then applied to align each ground-truth node with its most similar counterpart in the predicted set. Let $\mathcal{M}$ denote the matched node pairs and $\text{sim}(i, j)$ their cosine similarity. The accuracy reward is computed as:

$$\text{accuracy\_reward}_{\text{UML}} = \frac{1}{|\mathcal{M}|} \sum_{(i,j) \in \mathcal{M}} \text{sim}(i, j) \tag{6}$$

Each node is treated as an equally weighted evaluation unit, and the final reward reflects the average semantic fidelity of matched activity steps across all partitions.

(B) TEXT-BASED EVALUATION. For text-only cleaning plans, we compute similarity at the document level. The predicted and reference plans are treated as entire paragraphs and embedded as a whole using the same encoder (`all-MiniLM-L12-v2`).

Let $v^{\text{pred}}$ and $v^{\text{ref}}$ denote the embedding vectors of the predicted and reference cleaning plans, respectively. The accuracy reward is then defined as:

$$\text{accuracy\_reward}_{\text{text}} = \cos(v^{\text{pred}}, v^{\text{ref}}) \tag{7}$$

This formulation avoids sentence segmentation and captures holistic semantic similarity between the entire predicted and ground-truth plans.

(C) SUMMARY. The final accuracy reward is selected as:

$$\text{accuracy\_reward} = \begin{cases} \text{accuracy\_reward}_{\text{UML}} & \text{UML-formatted} \\ \text{accuracy\_reward}_{\text{text}} & \text{Textual} \end{cases}$$

This dual-mode reward computation allows consistent evaluation across symbolic and natural language output styles, enabling flexible fine-tuning strategies.

## C  FULL TRAINING ARGUMENTS BY STAGE

For reproducibility and clarity, table 4provide the complete training arguments used in each stage of our pipeline. Stage 1 corresponds to supervised fine-tuning (SFT) on CoT-labeled data, Stage 2 applies GRPO on the same CoT-labeled subset with customized reward functions, and Stage 3 continues GRPO on the answer-only subset.

Table 4: Key configurations across the three training stages.

| Setting | Stage 1: SFT | Stage 2: GRPO (CoT Data) | Stage 3: GRPO (Answer-Only) |
|---|---|---|---|
| Data | CoT-labeled JSON | CoT-labeled subset | Answer-only subset |
| Epochs | 5 | 3 | 2 |
| Batch size / Accum. | 4 / 4 | 8 / 2 | 8 / 2 |
| Precision | bf16 | bf16 | bf16 |
| Frozen modules | Backbone frozen | None frozen | None frozen |
| Max seq. length | 4096 | 4096 | 4096 |
| Learning rate | 4e-5 (cosine) | — (policy gradient) | — (policy gradient) |
| Reward funcs | — | Accuracy, Format | Accuracy, Format |
| Reward method | — | Clean_plan_UML | Clean_plan_UML |
| Generations | — | 8 | 8 |
| Beta | — | 0.04 | 0.04 |
| Deepspeed | ZeRO-1 | ZeRO-2 | ZeRO-2 |

# D EXTENDED UML DIAGRAM EXAMPLES

We present three representative examples from the MRoom-30k dataset. For each example, we display: i) A real-world messy room image input; ii) The generated UML class diagram representing structured chain-of-thought (CoT) reasoning; and iii) The corresponding UML activity diagram encoding an executable cleaning plan.

These cases demonstrate the model's ability to translate complex spatial messes into interpretable symbolic reasoning and structured, actionable cleaning strategies.

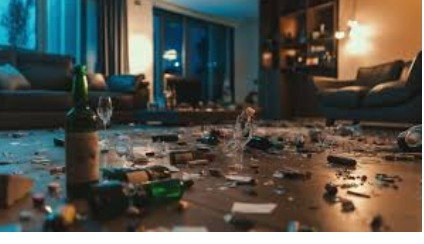

## D.1 EXAMPLE
1: LIVING ROOM WITH BROKEN GLASS AND TRASH

The image (Figure 10) depicts a cluttered living space with broken glass, scattered bottles, and floor-level hazards. The model identifies key mess areas and generates a safety-first cleaning strategy.

Figure 10: Input image (Example 00254): A heavily cluttered living room with bottles, glass shards, and surface debris.

As illustrated in Figures 11a and 11b, the structured diagrams represent mess categorization, prioritization, and modular cleanup logic.

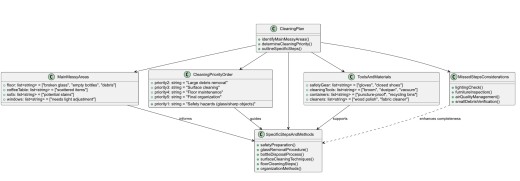

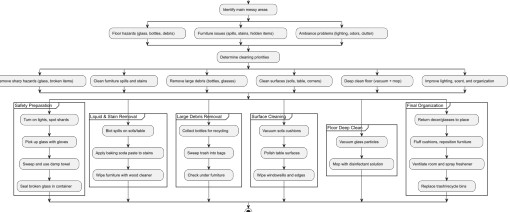

(a) UML Class Diagram: The structured CoT identifies four main messy zones (floor, coffee table, sofa, windows), categorizes mess types (e.g., "broken glass", "scattered items"), and links them with corresponding cleaning priorities and tools. The diagram includes fallback modules such as missed step considerations for robustness.

(b) UML Activity Diagram: The plan first addresses hazardous items (glass shards), then proceeds through logical cleaning phases: bottle disposal, surface wiping, floor vacuuming, and final reorganization. Specific methods are modularized for safety, efficiency, and completeness.

Figure 11: Structured reasoning and executable plan generated for an image with broken glass and cluttered zones.

## D.2 EXAMPLE 2: STORAGE
### ROOM WITH MIXED TOOLS AND DISORGANIZATION

This example (Figure 12) shows a cluttered study room with cardboard boxes, scattered tools, and multiple organization challenges. The model outputs a modular cleanup workflow that balances safety and order.

As shown in Figures 14a and 14b, the diagrams reflect object-oriented reasoning and compositional task plans.

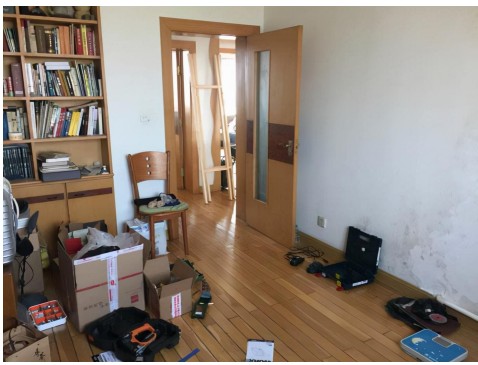

Figure 12: Input image (Example 00877): A disorganized study room with cardboard boxes, tools, books, and wall stains.

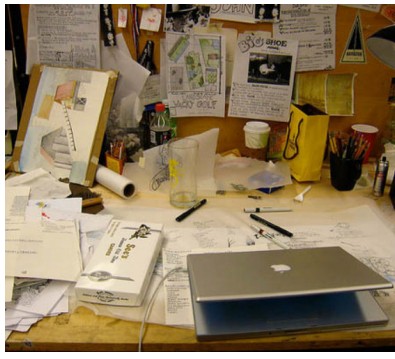

Figure 13: Input image (Example 01123): A cluttered work desk with papers, electronics, and mixed stationery.

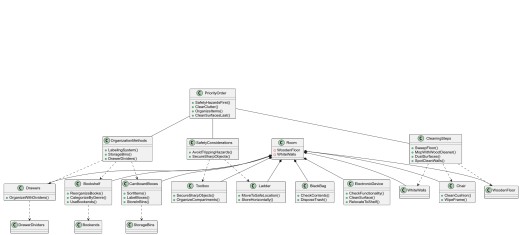

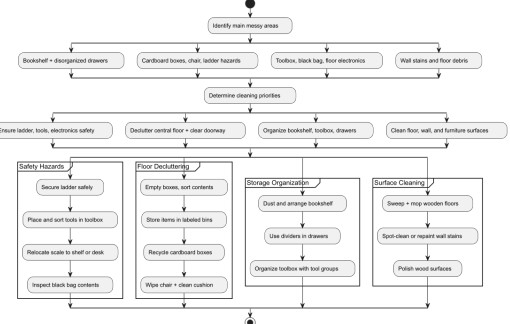

(a) UML Class Diagram: The reasoning process organizes the room via object-centric modules like bookshelf, toolbox, black bag, etc. Each object is tied to relevant actions, organized hierarchically under cleaning priorities and safety considerations. The structure facilitates task grouping and dependency handling.

(b) UML Activity Diagram: The plan starts with safety (securing ladder and tools), then performs floor decluttering (sorting, binning), followed by object-specific storage (foam inserts, dividers), and ends with surface polishing. The sequence ensures both hazard reduction and semantic order.

Figure 14: Structured reasoning and execution plan for a cluttered workshop scene with tool and storage elements.

## D.3 EXAMPLE 3: DESK SCENE WITH PAPER CLUTTER

This case (Figure 13) involves a messy work desk with scattered documents, electronics, and writing tools. The model infers a structured prioritization strategy across semantic object groups.

The reasoning and planning structures, shown in Figures 15a and 15b, highlight the use of high-/medium/low task tiers.

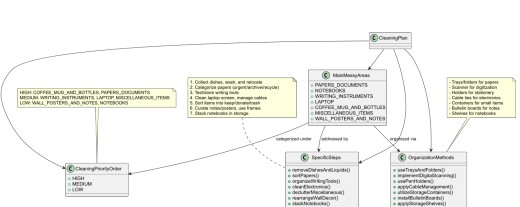

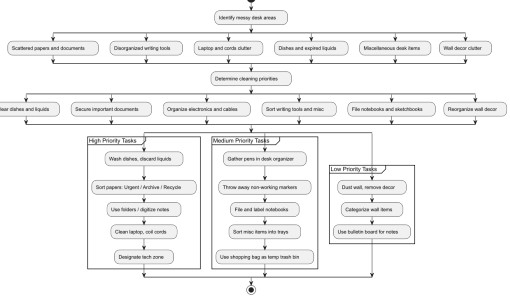

(a) UML Class Diagram: The CoT decomposes the desk scene into semantically meaningful object groups: papers, writing tools, electronics, wall decor. Cleaning priorities (high/medium/low) are assigned based on object criticality and mess severity. Organization methods (e.g., trays, cables) are explicitly encoded.

(b) UML Activity Diagram: The cleaning plan is split into three priority levels. High-priority actions (washing, paper sorting) are followed by medium (tool gathering, notebook filing), then low (dusting, wall organization). Task modularity reflects practical execution flows.

Figure 15: Structured CoT and executable plan for a desk-centric messy scene.

