# OpenReview forum: "UML-CoT: Structured Reasoning and Planning with Unified Modeling Language for Robotic Room Cleaning"
_ICLR.cc/2026/Conference — ICLR 2026 Conference Withdrawn Submission_

### Official Review · Reviewer_xG9S · 2025-10-18

**Soundness:** 2
**Presentation:** 3
**Contribution:** 2
**Rating:** 2
**Confidence:** 4

**Summary:**

# Summary
## What is the problem solved? Is it a known problem?

The problem that is being tackled in this work is plan generation for the robotics room cleaning task, but the restriction to robotics room cleaning can be alleviated and one can say the paper deals with plan generation in hierarchically represented domains.

## How is it solved in the literature and how is the current approach different?

The authors propose to use UML to describe both the planning problem and the hierarchical solution. This is not a common approach to planning problems, but it was tried in the past, with the work dating back about 20 years, e.g., [1,2,3]. To the best of my knowledge, the line of research was abandoned, possibly in favor of other, more convenient formalisms that allow capturing hierarchies and partial solutions, such as HTN planning [4,5]. I would encourage the authors to look at HDDL, which seems to allow capturing the aspects of the planning problems they explore in this work.

[1] Tiago Stegun Vaquero et. al., ICAPS 2005, The itSIMPLE tool for Modeling Planning Domains
[2] Tiago Stegun Vaquero et. al., ICAPS 2006, On the Use of UML.P for Modeling a Real Application as a Planning Problem
[3] Tiago Stegun Vaquero et. al., ICAPS 2007, itSIMPLE2.0: An Integrated Tool for Designing Planning Domains
[4] Erol et al., AAAI 1994, HTN planning: complexity and expressivity
[5] Holler et al., AAAI 2020, HDDL: An Extension to PDDL for Expressing Hierarchical Planning Problems


# Significance

The problem of automated plan generation is one of the major problems autonomous agents are tasked with. The currect solution proposed might not be adopted by the community, probably due to the same reasons the similar solution that was proposed 20 years ago was not adopted - it is quite complicated and even small errors in the captured/learned model can render the solution invalid. Also, the previously proposed solution proposed UML as a representation language, while proposing transformations into PDDL or petri nets, for which solvers are readily available. Here, the authors propose language models to do the solving, aided by SFT.

# Soundness

There is no theory presented in the paper.

# Novelty

As mentioned before, the novelty is somewhat limited.

# Scholarship

The previous work on modeling with UML or on hierarchical planning are not mentioned (see some representative papers above). HTN planning is an area of active research for the past 45+ years, with thousands of papers, workshop series and competitions dedicated to the topic.

# Clarity

The paper is rather well written, with the ideas clearly described.

# Evaluation and Reproducibility

The experiments are performed on the MRoom-30k dataset, in distribution only, using the 80/10/10 split. The evaluation is performed based on semantic similarity using a language model. It is unclear therefore how the approach performs (as the evaluation is imprecise) and whether it generalized out of distribution (not tested).

The baselines are rather weak, CoT/ToT/GoT. The ToT/GoT in addition to being extremely inefficient (hundreds of calls to the language model to solve each task) were recently also shown to exhibit severe performance degradation when moving out of the distribution of the instances the model was initially trained on (see Table 1 bottom part in [6]). Still, even compared to these weak baselines the performance of the proposed approach is essentially the same. Ablations show that without finetuning the performance is very low, so the finetuning is crucial to the success of the approach. It is not clear therefore whether the finetuned model doesn't just memorize the "correct" (according to semantic similarity evaluation metric) answers.

[6] Katz et al., Arxiv 2025, Seemingly Simple Planning Problems are Computationally Challenging: The Countdown Game

**Strengths:**

1. The paper is nicely written
2. The problem is clearly described (albeit not formally defined)
3. The approach is presented with visual aids, diagrams and UML example problem and solution description

**Weaknesses:**

1. The paper on planning ignores the relevant planning literature (well, any planning literature).
2. There is no motivation for why UML would be a good formalism to use with language models.
3. There is no formal definition of what the problem is (what is the input, what is considered to be a solution).
4. The authors do not provide a sound validator for their task, rendering the experiments uninformative.
5. The experiments do not investigate out of distribution performance.
6. The experiments do not compare to strong baselines (e.g., HTN planners).

**Questions:**

1. Can all aspects of your problem be captured by HTN planning and in particular by HDDL? If not, what aspects cannot be captured?
2. How do you ensure that the UML problem description generated is correct?
3. How would you validate generated solutions? The semantic similarity does not answer whether the produced output is an actual plan.

---

### Official Review · Reviewer_o6Xi · 2025-10-26

**Soundness:** 3
**Presentation:** 3
**Contribution:** 2
**Rating:** 2
**Confidence:** 4

**Summary:**

The paper proposes UML-CoT, representing symbolic reasoning as UML class diagrams and executable plans as UML activity diagrams, trained with a three-stage pipeline: SFT on UML traces, RL fine-tuning using final-plan rewards via GRPO, and GRPO on answer-only data.​
A new MRoom-30k dataset of messy rooms is constructed with final plans generated by GPT-4o and a 1k subset with CoT from DeepSeek-R1, and evaluation relies on a semantic-similarity reward using all-MiniLM-L12-v2 over partitioned plan sections with greedy matching, plus a format reward for tag correctness.​ Experiments report that fully UML-based reasoning and plans, especially with the extra RL stage, improve similarity, recall, and F1 over text CoT, Tree-of-Thoughts, and Graph-of-Thoughts on a 1k test subset, with ablations showing SFT is necessary to bootstrap valid UML outputs and GRPO adds gains beyond longer SFT.​ Cross-task generalization results for cooking and painting are reported but under-specified, and there is no closed-loop execution in simulators or on hardware, leaving executability claims supported only by text similarity proxies rather than task success.​

**Strengths:**

1. Novel formalism: Leveraging UML for symbolic reasoning and planning is creative and theoretically grounded in software engineering principles.

2. Structured interpretability: The approach enhances transparency and debuggability compared to unstructured CoTs or scene graphs.

3. Methodological rigor: The three-stage learning pipeline (SFT - RLFT - GRPO) is well-motivated and demonstrates incremental gains.

4. Empirical evaluation: Results are consistent across multiple baselines (Tree/Graph of Thoughts) and show measurable improvements in semantic similarity and recall.

5. Dataset contribution: The introduction of MRoom-30k for messy room reasoning fills a niche gap in embodied AI benchmarks.

**Weaknesses:**

1. Rewards are computed only from the final plan with an embedding similarity and a format bonus, providing no direct supervision for intermediate class-diagram quality and inviting reward hacking or spurious similarity rather than faithful stepwise reasoning and grounding.

2. The accuracy metric depends on all-MiniLM-L12-v2 cosine similarity and a bespoke partitioning and greedy matching procedure, but there is no human preference or simulator/hardware execution study to show that higher similarity truly correlates with cleaning success or safety.

3. Cross-task generalization lacks a clear protocol: training data, zero-shot versus fine-tuned settings, annotation sources, schema alignment, and overlap controls are missing, which makes the cooking and painting results in Table 2 hard to interpret causally.

4. The paper does not benchmark on established embodied household datasets such as ALFRED, TEACh, or Habitat Rearrangement, which provide standard execution-based metrics and would substantiate claims about planning quality and generality in interactive settings.

5. The case for UML versus typed JSON schemas or DSL-based plans, function-calling tool APIs, or PDDL is purely conceptual without controlled ablations comparing these structured alternatives on identical backbones and metrics. (The paper lacks clear comparison to simpler structured prompts (e.g., JSON or DSL-based plans....) that might offer similar interpretability with less formal complexity.)

6. There is no hybrid grounding pipeline leveraging scene graph extraction to populate UML class diagrams from perception, weakening claims about improved world modeling and verifiability in visual scenes.

 (Minor weakness)

7. Dataset labels are largely LLM-generated (GPT-4o and DeepSeek-R1), and evaluation converts textual outputs to UML via GPT-4o, which risks label noise, conversion bias, and metric circularity disconnected from actual robotic success.

8. Claims of “executable plans” are not backed by execution in ALFRED, TEACh, Habitat (ReplicaCAD/HSSD), or real robots such as TidyBot-like settings, which would give stronger external validity for executability beyond similarity scores.

9. UML modeling introduces heavy symbolic overhead; it’s unclear how the approach scales to dynamic or continuous control tasks.

**Questions:**

1. How exactly is cross-task generalization configured: are cooking and painting strictly zero-shot from room-cleaning training, or is any adaptation performed, and what are the data sources, schemas, and overlap controls for those domains ?

2. Can you provide controlled ablations comparing UML activity/class diagrams to a typed JSON schema with control-flow operators, a function-calling interface, and a PDDL pipeline on the same training/evaluation setup to substantiate the choice of UML ?

3. What mechanisms ensure the class diagrams are accurate and consistent with the activity diagrams if the reward only scores the final plan, and do you compute any structural validity or cross-diagram consistency constraints beyond format checking ?

4. Why not ground UML via a perception-first pipeline that extracts a scene graph and lifts it into UML programmatically to reduce hallucination and improve consistency, and how would that compare to direct UML generation? YOur intution.

5. (optional) Can you validate the similarity metric with human ratings and execution-based metrics in ALFRED or TEACh to demonstrate that higher similarity corresponds to higher success rates and better path efficiency in embodied tasks ?

6. If the ultimate goal is rearrangement/cleanup, why not include benchmarks on Habitat Rearrangement (with ReplicaCAD/HSSD scenes) to measure object-goal success and state deltas under physics, and explain any interface incompatibilities if omitted ?

7. How sensitive are results to the predefined partitions (“Main Messy Areas,” “Priority,” “Steps”), and are these partitions domain-neutral enough to evaluate cooking/painting without bias or misalignment ? IN case I missed this in paper, please point it out.

---

### Official Review · Reviewer_rR63 · 2025-10-31

**Soundness:** 1
**Presentation:** 2
**Contribution:** 1
**Rating:** 2
**Confidence:** 4

**Summary:**

The paper introduces a structured reasoning framework that models both reasoning and planning as UML diagrams. Using a custom GPT-4o-annotated dataset (MRoom-30k), the authors train the model through supervised fine-tuning on a small CoT subset and two stages of GRPO reinforcement learning. The method aims to enhance structure adherence and interpretability, showing improved results over textual reasoning baselines.

**Strengths:**

- The topic is engaging and the research direction of structured and embodied reasoning is practical.

- The formulation of reasoning as the generation of UML diagrams is an interesting attempt.

**Weaknesses:**

- The proposed method primarily targets the room-cleaning domain, raising concerns about its generalizability. In Table 2, the Cooking and Painting tasks are reported without clarifying the source. Moreover, all experiments are conducted solely on a custom dataset (MRoom-30k). It would strengthen the work to include evaluations on public benchmarks.

- The quality of the custom dataset is uncertain, as ground-truth plans are generated by GPT-4o without verification. Given that the labels are in symbolic representations, executing the generated plans with an external simulator (see PlanBench [1]) could help validate their correctness.

- The learning signal design raises concerns. Only about 1k samples contain CoT annotations for SFT, which may be insufficient. During RL fine-tuning, rewards derived from graph alignment can be unreliable due to the inherent difficulty of matching predicted and reference graphs. Furthermore, the rationale for separating CoT RL training (Stage 2) from plan-level RL training (Stage 3) is unclear.

- The reasoning baselines are limited. Comparisons should include VLM agents that ground reasoning to perception and actions, such as ProgPrompt [2], OpenVLA [3], and Open-X Embodiment [4].

- The novelty appears limited, focusing mainly on dataset construction and prompt-format modifications.

- Code and data are not provided for review.

## Reference:
[1] Valmeekam, Karthik, et al. "Planbench: An extensible benchmark for evaluating large language models on planning and reasoning about change." Advances in Neural Information Processing Systems 36 (2023): 38975-38987.

[2] Singh, Ishika, et al. "Progprompt: Generating situated robot task plans using large language models." arXiv preprint arXiv:2209.11302 (2022).

[3] Kim, Moo Jin, et al. "Openvla: An open-source vision-language-action model." arXiv preprint arXiv:2406.09246 (2024).

[4] O’Neill, Abby, et al. "Open x-embodiment: Robotic learning datasets and rt-x models: Open x-embodiment collaboration 0." 2024 IEEE International Conference on Robotics and Automation (ICRA). IEEE, 2024.

**Questions:**

See weaknesses.

---

### Official Review · Reviewer_u88V · 2025-11-01

**Soundness:** 1
**Presentation:** 2
**Contribution:** 2
**Rating:** 4
**Confidence:** 4

**Summary:**

This paper introduces UML-CoT, a new structured chain-of-thought (CoT) framework that improves multimodal reasoning and planning for embodied robotic room-cleaning tasks. Traditional CoT prompting uses free-form natural language, which lacks explicit structure and can be ambiguous, hard to interpret, or inconsistent. In this paper, the proposed UML-CoT uses Unified Modeling Language (UML) to explicitly represent reasoning and planning. Specifically, UML Class Diagrams encode objects, attributes, relationships (inheritance, aggregation), and serve as symbolic Chain-of-Thought. Besides, UML Activity Diagrams represent executable plans and support sequencing, branching, loops. Together, they yield a more interpretable and executable reasoning pipeline. Across multiple evaluation settings, UML-CoT demonstrates higher plan correctness, more coherent sequential steps and improved interpretability.

**Strengths:**

1. Stronger Structure than Text-based CoT. Traditional free-form Chain-of-Thought is ambiguous, hard to verify and difficult to execute. In contrast, UML-CoT provides a standardized symbolic representation by using UML class diagrams (reasoning) and UML activity diagrams (planning), which enables consistent, interpretable, and modular CoT.

2. Better Expressiveness than Graph-based Methods. Prior structured CoTs (scene/logic graphs) lack inheritance, aggregation and behavioral abstraction. However, UML overcomes these issues by natively supporting object hierarchies, attributes + relationships and conditional / sequential / iterative flows. The proposed method provides more expressive symbolic modeling and planning.

3. New Dataset MRoom-30k. The paper develops a large, diverse messy-room dataset, which contains 30k+ real messy indoor images, cleaning plans and 1k reasoning traces. It enables controlled evaluation of structured reasoning.

**Weaknesses:**

Authors claim that the proposed method is domain-adaptable. However, all the evaluations are in a single domain of robotic room cleaning. The domain-adaptable capability of the proposed method is not evaluated comprehensively or convincingly enough.

1. The major concern of this work is its domain-adaptable capability. The agent model is pre-trained with a fixed dataset on room cleaning, and the training of UML generation is also limited to the room cleaning domain. The UML has its own language restrictions and is not flexible enough to easily adapt to changes of domains. Authors demonstrate the cross-task generalizability of the proposed method in the experiments. However, it is still about transferring to different tasks in the same domain, not related to domain-adaptability. The discussion is short and experiments are not comprehensive enough. So, the evaluation of domain-adaptability is not convincing enough.

2. Another weakness of the proposed method is that, as an agent model, its computation power and resources are limited in the real-world applications. If the model is too large or its fine-tuning is too expensive, it cannot be applicable to agentic applications. A small model for building the agent is preferred. It is better to avoid fine-tuning in adapting to changes of environment or working domain. However, the authors did not discuss these issues in the paper.

3. The training pipeline is complex and expensive. Three-stage training includes SFT, RLFT and GRPO. This pipeline is long and computationally intensive, which makes the reproducibility more difficult. Besides, the details of the training part are introduced comprehensively enough, and choices of important hyperparameters are also missing, which cannot make the performance of the proposed method convincing enough.

4. The baselines in the experiments are also limited. As an embodied AI paper, the baselines should include symbolic task planners (PDDL-based) and RL-based navigation methods. Its advantages over state-of-the-art planning methods are not clear.

**Questions:**

1. In the construction of the dataset, most images are annotated with cleaning plans generated by GPT-4o. However, it is widely known that GPT models can hallucinate and produce wrong plans which may violate some domain constraints. Wrong plans in the training data can make the agent work wrongly in the deployment. How does the proposed method address this issue?

2. In the RL training part, the accuracy reward is only the semantic similarity between generated and ground-truth plan. But semantic similarity may not accurately reflect the correctness of the generated plan. Sometimes, the generated plan may be semantically similar as the ground-truth result, but incorrect essentially. How does the proposed framework resolve this situation?

3. In the RL training part, the format rewards encourage the model to generate both <think> and <answer> tags. But both tags are required for the agent to work correctly based on UML. So, encouraging reward may not be strong enough to make sure the trained agent works properly. Why not penalize the model if any of these tags are missing?

---

### Note · Authors · 2025-11-12

I have read and agree with the venue's withdrawal policy on behalf of myself and my co-authors.